# Entomopathogenic Fungi as Dual Control Agents against Two Phytopathogens and the Lepidopteran Pest *Rachiplusia nu* in Soybean (*Glycine max* (L.) Merr)

**DOI:** 10.3390/jof10020093

**Published:** 2024-01-24

**Authors:** María Leticia Russo, María Florencia Vianna, Ana Clara Scorsetti, Natalia Ferreri, Juan Manuel de Abajo, María Inés Troncozo, Sebastián Alberto Pelizza

**Affiliations:** 1Instituto de Botánica Carlos Spegazzini, Facultad de Ciencias Naturales y Museo, Universidad Nacional de La Plata, Avenida 122 y 60, La Plata 1900, Argentina; russomaleticia@gmail.com (M.L.R.); anacscorsetti@yahoo.com (A.C.S.); ferrerinataliaanalia@gmail.com (N.F.); juanmdabajo@gmail.com (J.M.d.A.); mariainestroncozo@hotmail.com (M.I.T.); sebastianpelizza@yahoo.com.ar (S.A.P.); 2Consejo Nacional de Investigaciones Científicas y Técnicas, Godoy Cruz 2290, Ciudad Autónoma de Buenos Aires 1425, Argentina; 3Comisión de Investigaciones Científicas de la Provincia de Buenos Aires (CICPBA), Calle 526 e/ 10 y 11, La Plata 1900, Argentina

**Keywords:** antifungal activity, biological control, entomopathogenic fungi, phytopathogenic fungi, *Rachiplusia nu*

## Abstract

Soybean (*Glycine max*) is one of the most important crops worldwide. This crop is prone to diseases caused by the phytopathogenic fungi *Macrophomina phaseolina*, *Fusarium oxysporum,* and the lepidopteran pest *Rachiplusia nu*. Biological control using entomopathogenic fungi is a sustainable alternative to chemical pesticides. In this study, we assessed the antifungal activity of *Beauveria bassiana* and *Metarhizium anisopliae* against phytopathogens and the pathogenicity of *B. bassiana* as an endophyte against *R. nu*. The antagonic activity of the fungal entomopathogens was evaluated in vitro by dual culture bioassays. The feeding preference of *R. nu* was evaluated in a “free choice” assay under laboratory conditions. Entomopathogenic fungi inhibited the mycelial growth of the phytopathogens. The best response in all cases was observed when the antagonists were placed in the culture medium two days before the pathogens. *B. bassiana* was the best antagonist of *F. oxysporum*, while both antagonists had similar inhibitory effects on *M. phaseolina* growth. Additionally, *B. bassiana*, when grown as an endophyte, reduced insects’ food preferences and decreased soybean consumption. Findings suggest that entomopathogenic fungi can fulfill multiple roles in the ecosystem. Therefore, the simultaneous expression of both properties should be considered for their application in integrated pest management programs.

## 1. Introduction

Soybean (*Glycine max* L. Merr) is a crop with significant economic importance and one of the most valuable commodities globally [1]. It is highly adaptable to various environmental conditions, making it a highly sought-after crop worldwide, with an annual planting area of 120 million hectares and 333 million tons of harvest [2,3]. Argentina is the third-largest soybean producer and the leading oil and meal exporter.

During their growth cycle, soybean plants are vulnerable to damage from various arthropod pests and pathogenic fungi, which can result in significant economic losses [4,5].

Soybean plants provide vital resources for herbivores; thus, some severe pests negatively impact the grain yield of the crop [6,7]. *Rachiplusia nu* (Guenée) (Lepidoptera: Noctuidae: Plusiinae) is a defoliating pest of soybean crops in Argentina, southern Brazil, Paraguay, Uruguay, and Bolivia [8]. Its populations have increased in recent years and caused problems in the early vegetative and grain-forming stages of soybean crops. The early-stage larvae of *R. nu* feed on the leaf parenchyma, most voraciously during their fifth larval instar. The larval feeding decreases the leaf area, causing a lower photosynthetic capacity, loss of stored leaf material, and shortening of the grain filling period [9,10]. 

On the other hand, *Fusarium oxysporum* (Ascomycota: Hypocreales) and *Macrophomina phaseolina* (Ascomycota: Botryosphaeriales) are fungal pathogens that heavily infect soybean crops and cause diseases that are transmitted through soil and seeds. More serious infections can reduce yield and seed quality and lead to pod abortion [5,11,12]. *Fusarium oxysporum* is one of the most destructive fungi that can cause root rot and wilt in soybeans [13]. Infected roots become shallow, fibrous, and eventually rot. Plants can wilt, especially in low humidity and high temperatures [14,15].

*Macrophomina phaseolina* is an economically important necrotrophic fungus that infects soybean crops at any growth stage, from seedlings to mature plants [16]. The disease is known as “charcoal rot” and causes root rot and rapid wilting, keeping the leaves attached [17]. Chemical insecticides and fungicides have become the control methods most widely used in agriculture against pest arthropods and severe fungal diseases that limit crop growth and yield [18]. However, although they have proven effective in various cases, they have also brought new and unforeseen issues, such as human toxicity and ecosystem imbalance due to the decrease in abundance and diversity of non-target organisms [8,9,19,20]. Consequently, the search for more targeted and environmentally friendly control methods is of particular interest [8,21]. The entomopathogenic fungi *Beauveria bassiana* (Ascomycota: Hypocreales) and *Metarhizium anisopliae* (Ascomycota: Hypocreales) are natural enemies of many insects and have been extensively studied for biological control [22,23,24]. As endophytes, they have been inoculated in different plant species through various techniques [25]. Several genera of entomopathogenic fungi have been shown to protect plants against fungal pathogens. Studies have demonstrated that *B. bassiana* is effective against *Fusarium* species [14,26,27,28] and *M. anisopliae* against *F. oxysporum* [27,29] and *F. graminearum* [30]. Nevertheless, the antagonistic effects of *M. anisopliae* and *B. bassiana* against *M. phaseolina* have not been previously evaluated. On the other hand, previous studies have shown the capacity of *M. anisopliae* and *B. bassiana* to establish as endophytes in soybean plants and control insect pests [31,32].

Our study aimed to evaluate the antifungal activity of two entomopathogenic fungi, *B. bassiana* and *M. anisopliae*, against *F. oxysporum* and *M. phaseolina.* We also examined the endophytic capacity of *B. bassiana* for controlling *R. nu* to comprehensively determine their ability as dual biological controllers of pest insects and phytopathogenic fungi. 

## 2. Material and Methods

*Beauveria bassiana* LPSc 1098 (accession number KT163259) was isolated from *Triatoma infestans* (Hemiptera: Reduviidae) from Chaco province, Argentina, and *Metarhizium anisopliae* LPSc 907 (accession number KJ772494) was isolated from Hemiptera: Cercopidae from Buenos Aires province, Argentina. Both were obtained from the culture collection of the “Instituto de Botánica Carlos Spegazzini” (LPSc). *Fusarium oxysporum* LPSc 1191 (accession number KF753954) and *Macrophomina phaseolina* LPSc 1185 (accession number KF753945) were isolated as endophytes from soybean crops in the Pampean Region of Argentina. These strains were also deposited in the culture collection and identified morphologically and molecularly by Pelizza et al. [33] and Russo et al. [34].

All strains were grown on potato glucose agar (PDA: Britania S.A., Buenos Aires, Argentina) at 25 °C, 85% RH, for 7–10 days in darkness and stored at 4 °C until use.

### 2.1. Bioassay I: In Vitro Antagonism (Dual Culture Assays)

The antagonistic activity of *M. anisopliae* and *B. bassiana* against *F. oxysporum* was evaluated in vitro. For each fungal isolate, mycelium discs of 5mm were obtained from seven-day-old pure cultures grown on potato dextrose agar (PDA) in Petri dishes [35]. One fungal antagonist (*M. anisopliae* or *B. bassiana)* was tested with one fungal pathogen (*F. oxysporum* or *M. phaseolina*) by inoculating the antagonist and pathogen onto opposite sides of 90-mm Petri dishes with 15 mL of PDA. Three treatments were used [36]: (a) treatment I: pathogens and antagonists were inoculated simultaneously; (b) treatment II: the pathogens were inoculated two days before the antagonists; and (c) treatment III: the antagonists were inoculated two days before the pathogens. Control plates contained only the antagonist or pathogen mycelial disc to check growth without interactions. The plates were incubated at 24 °C, 85% RH, in darkness for ten days [29], and the radial growth of the pathogens was measured. All the experiments were carried out in five replicates and repeated twice.

The radial growth percentage inhibition of the phytopathogenic fungi compared to control growth was calculated using the formula according to Barra-Bucarei et al. [19]: [(R1 − R2)/R1] × 100

R1: radius of the pathogen colony in the control;

R2: radius of the pathogen colony in the interaction.

### 2.2. Bioassay II: Consumption and Feeding Preference

#### 2.2.1. Insect Breeding

*Rachiplusia nu* was artificially reared in chambers maintained under controlled conditions of temperature (25 ± 2 °C), relative humidity (70–75%), and photoperiod (14:10 h L:D) at the “Instituto de Botánica Carlos Spegazzini” following Barrionuevo et al. [37]. The artificial diet used to feed the insects was based on Osores et al. [38]. To start the colony, the eggs were obtained from AgIdea S.A. (www.agidea.com.ar, accessed on 20 November 2022) [39] to avoid contamination by field insects.

#### 2.2.2. Obtaining and Inoculating Soybean Plants with *B. bassiana*

Soybean seeds of the DM 3810 variety (DM Catalogo–DonMario Semillas) were used. The seeds were superficially sterilized and sown in 300 cm^3^ pots with a sterile substrate consisting of a mixture of soil, perlite, and vermiculite in equal parts (1:1:1), following Russo et al. [7]. The plants were inoculated using foliar spraying, the most effective technique for introducing these strains. Inoculation was performed when the plants were ten days old and had developed true leaves. A hand sprayer was used to spray the abaxial surface with 2 mL of conidial suspension at a concentration of 1 × 10^8^ conidia/mL per plant [40]. Control plants were sprayed with 2 mL of a 0.01% (*v*/*v*) Tween 80 sterile water solution free of conidia [25,41,42]. Conidial concentration was determined using a Neubauer chamber. Conidial viability was assessed according to Greenfield et al. [43]. In all cases, the mean conidial viability was >95%.

#### 2.2.3. Consumption and Feeding Preference

The “free choice method” was used to assess the feeding preference of *R. nu* [25]. *Rachiplusia nu* L2 larvae were fasted for 4 h and then individually placed in 9 cm diameter Petri dishes with moistened filter paper. Soybean leaf discs (3 cm diam.) previously weighed, one inoculated with *B. bassiana* (seven days post-inoculation) and another from control plants, were simultaneously offered to the larvae. The plates were incubated at 25 °C, 60% relative humidity, and a 14:10 h LD photoperiod for 24 h, allowing the larvae to choose between both discs. Three repetitions of 30 individuals each were performed. After 24 h, the fresh weight of the uneaten disc was obtained, and the remains were oven-dried at 60 °C until a constant weight was obtained. The ingested food was obtained from the difference between the dry weight of the disc offered and the remaining material at the end of the experiment. The leaf’s initial dry weight was estimated from the initial fresh weight using a correction factor obtained from the average dry weight/fresh weight ratio of a sample of control leaves. Dry weights were used due to the great variability in the leaf water content. The difference in weight between the offered discs and the remaining material represented the consumption during the assay [44,45]. 

### 2.3. Data Analysis

The antagonistic effects of the entomopathogenic fungi *B. bassiana* and *M. anisopliae* on the growth of *F. oxysporum* and *M. phaseolina* were analyzed using a two-way analysis of variance (ANOVA) (independent variable: strains of entomopathogenic fungi and pairing method (treatments) and response variable: radial growth inhibition of each phytopathogen (%)). A Tukey test was performed to explore the differences between the groups (*p* < 0.05). The effects of the LPSc 1098 strain on the feeding preference of *R. nu* larvae were investigated using a T-test (*p* < 0.05). All statistical analyses were performed with Infostat [46].

## 3. Results

### 3.1. Bioassay I: Antagonistic Activity of Entomopathogenic Fungi

The two entomopathogenic fungi inhibited *F. oxysporum* and *M. phaseolina* mycelial growth in the dual culture assay. Each pathogenic fungus differed significantly between both antagonists and between treatments (Table 1). 

In all treatments, control cultures of both pathogenic fungi completely colonized the Petri dishes after ten days (Figure 1). 

However, in the presence of antagonists, the colony diameter of the pathogens significantly decreased between treatments. Figure 2 and Figure 3 show the effects of *B*. *bassiana* and *M*. *anisopliae* on *F*. *oxysporum* and *M*. *phaseolina* growth, respectively.

Treatment III was the most effective in all cases; highly significant differences were found between treatment III and treatments I and II. Furthermore, treatments I and II did not differ significantly (Figure 4).

In treatment III, *B. bassiana* was the most effective antagonist against *F. oxysporum,* with 64.18% inhibition, while *M. anisopliae* inhibited 57.5%. On the contrary, antagonist inhibition did not exceed 50% in treatments I and II (Figure 4a). On the other hand, both *B. bassiana* and *M. anisopliae* were equally effective in inhibiting *M. phaseolina* growth in treatment III without showing significant differences (62% and 63% inhibition, respectively) (Figure 4b). Furthermore, when the growth of *M. phaseolina* was evaluated in the presence of *B. bassiana* and *M. anisopliae*, no significant differences were found between treatments I and II (Figure 4b). 

### 3.2. Bioassay II: Consumption and Feeding Preference

The foliar consumption of the L2 larvae of *R. nu* shows significant differences (T = 3.83, df = 58, *p* < 0.0008). *Beauveria bassiana* LPSc 1098 as an endophyte in soybean plants decreased insect feeding preference and soybean consumption.

Foliar consumption was lower in plants inoculated with the fungus (30–39 mg/insect) than in control plants (35–50 mg/insect), as shown in Figure 5. Thus, colonization and persistence of *B. bassiana* LPSc 1098 as an endophyte in soybean plants could decrease the feeding preference and, therefore, its consumption by the insect.

## 4. Discussion

Biological control of plant pathogens and insect pests is crucial to decrease reliance on chemical pesticides and fungicides and increase agricultural sustainability. In this study, we investigated the antimicrobial activity of the entomopathogenic fungi *B. bassiana* and *M. anisopliae* against two fungal pathogens of soybean plants, *F. oxysporum* and *M. phaseolina*. Also, we assessed the feeding preference of *R. nu*, a lepidopteran pest of soybean crops.

Some species of entomopathogenic fungi, such as *B. bassiana*, *Metarhizium brunneum*, *M. anisopliae*, *Lecanicillium lecanii*, and *Isaria javanica*, are effective in controlling certain phytopathogenic fungi and stramenopiles, including *Rhizoctonia solani*, *Pythium myriotylum*, *Sphaerotheca fuliginea*, *Botrytis cinerea*, *Fusarium oxysporum*, *Colletotrichium*, *Phytophthora*, and *Plasmopara viticola* [19,26,27,31,47,48,49,50,51]. As far as we are concerned, this is the first study to analyze the antagonistic effects of *M. anisopliae* and *B. bassiana* on *M. phaseolina*. Most control agents are only active against either insect pests or plant pathogens. However, our results indicate that *B. bassiana* exhibits a dual antagonistic effect. This is consistent with previous research, which has also observed the dual action of this fungus against *Botrytis cinerea, Alternaria alternata*, the aphid *Macrosiphum euphorbiae* [51], *B. cinerea*, and the aphid tomato pest *Myzus persicae* [52]. In all treatments, the antagonism of *B. bassiana* against *F. oxysporum* and *M. phaseolina* was effective. However, treatment III shows a better response, with an inhibition percentage of 64.18% and 62%, respectively. Culebro-Ricardi et al. [26] also observed a slower growth of *F. oxysporum* when *B. bassiana* was applied to the medium two days before the pathogen, reaching a 70% inhibition percentage, compared to simultaneous inoculation. Shternshis et al. [28] demonstrated that the antifungal activity of *B. bassiana* increased over time, as the diameter of the phytopathogen colony under the influence of *B. bassiana* decreased significantly. Jaber and Alananbeh [27] found that *B. bassiana* prevented the mycelial growth of three *Fusarium* species (*F. oxysporum, F. culmorum*, and *F. moniliforme*), with percentages lower than 62%. Our results also reveal that *M. anisopliae* behaved similarly to *B. bassiana* and responded better to *F. oxysporum* (57.5%) and *M. phaseolina* (63%) in treatment III. Likewise, previous studies have shown that *M. brunneum* prevented the mycelial growth of three *Fusarium* species (*F. oxysporum*, *F. culmorum,* and *F. moniliforme*), with percentages ranging between 48% and 57% [27]. On the other hand, Picardal et al. [29] observed that *M. anisopliae* moderately inhibited the in vitro radial growth of the pathogen (31.27%). Hao et al. [30] concluded that this fungus produces a clear inhibition zone in front of the *F. graminearum* colony, leading to observable deformation and branching of the pathogen hyphae. 

Several authors have considered that entomopathogenic fungi inhibit pathogen growth and development due to resource competition, antibiosis, and parasitism. The production of volatile compounds and bioactive secondary metabolites with antimicrobial properties are the mechanisms of action involved in these activities [53]. Our results are consistent with Jaber and Alananbeh [27], who provided evidence of clear inhibition zones between antagonists and pathogens. Therefore, it is likely that these antagonism zones are caused by inhibitory metabolites (antibiosis) produced by fungi. *Beauveria bassiana* produces a variety of important metabolites, including bassianolide, bassianin, beauveriolide, bassiacridine, cyclosporine, oosporein, and beauvericin, the last two with antifungal activity. *Metarhizium* sp. can produce other secondary metabolites, such as destruxins, swainsonines, serinocyclins, and cytochalasins [54]. The inoculation and establishment of entomopathogenic fungi in plants emerge as alternative and promising avenues to overcome the constraints of environmental conditions in pest control [55]. 

Our findings reveal that when given the option to choose between plants inoculated and not inoculated with the entomopathogenic fungus *B. bassiana*, *R. nu* larvae had a greater preference for control plants. Previous studies [25,32,56] also showed that *R. nu* and *Spodoptera frugiperda* larvae consumed fewer plants when the entomopathogenic fungus was present as an endophyte. Similarly, Castillo López et al. [57] and Martinuz et al. [58] confirmed through preference tests that *Aphis gossypii* preferred to feed on uncolonized plants. Vianna et al. [59] did not observe significant differences between the consumption of inoculated and non-inoculated tobacco plants by adults of *D. speciosa*, contrary to previous findings. Our results indicate that when given a choice, larvae tend to choose uncolonized leaves over those colonized by *B. bassiana* (evidenced by the significant reduction in the larval-preferred leaf weight). As a result, the soybean colonization by *B. bassiana* negatively impacted *R. nu* feeding preferences. According to Ownley et al. [31] and Russo et al. [25], this could be attributed to the antiherbivore properties of *B. bassiana*’s secondary metabolites in plants.

## 5. Conclusions

Entomopathogenic fungi have shown promising results regarding their antifungal activity and antiherbivore effects. This study highlights the multiple ecological roles that these fungi can play, further supporting their potential as control agents. It is crucial to consider the simultaneous expression of these fungi’s insecticidal and antifungal properties when considering their application in integrated pest management programs, particularly within the framework of sustainable agriculture.

## Figures and Tables

**Figure 1 jof-10-00093-f001:**
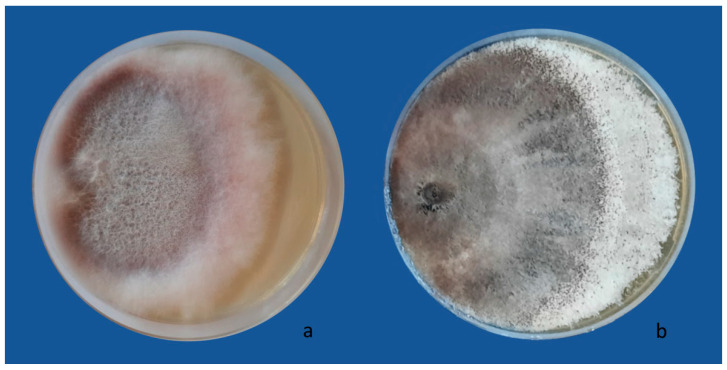
Radial growth of the phytopathogenic fungi *Fusarium oxysporum* (**a**) and *Macrophomina phaseolina* (**b**) in control plates.

**Figure 2 jof-10-00093-f002:**
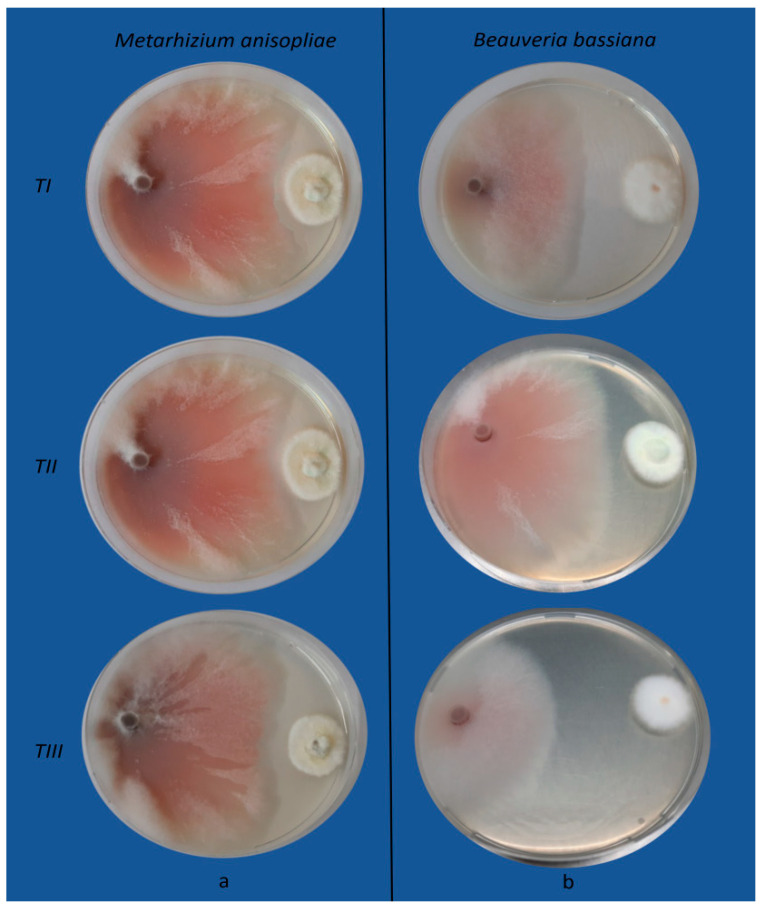
Antagonistic effects of *Metarhizium anisopliae* (**a**) and *Beauveria bassiana* (**b**) towards *Fusarium oxysporum* by the dual culture method. *TI*: the pathogen and the antagonists were inoculated simultaneously; *TII*: the pathogen was inoculated two days before the antagonists; *TIII*: the antagonists were inoculated two days before the pathogen.

**Figure 3 jof-10-00093-f003:**
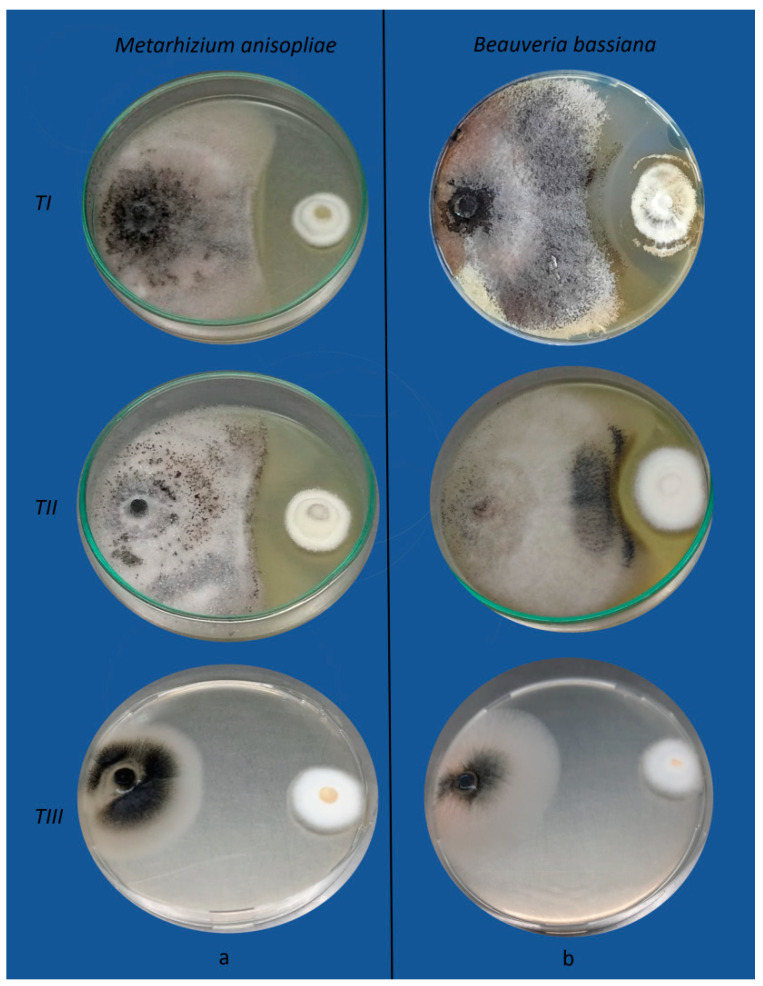
Antagonistic effects of *Metarhizium anisopliae* (**a**) and *Beauveria bassiana* (**b**) towards *Macrophomina phaseolina* by the dual culture method. *TI*: the pathogen and the antagonists were inoculated simultaneously; *TII*: the pathogen was inoculated two days before the antagonists; *TIII*: the antagonists were inoculated two days before the pathogen.

**Figure 4 jof-10-00093-f004:**
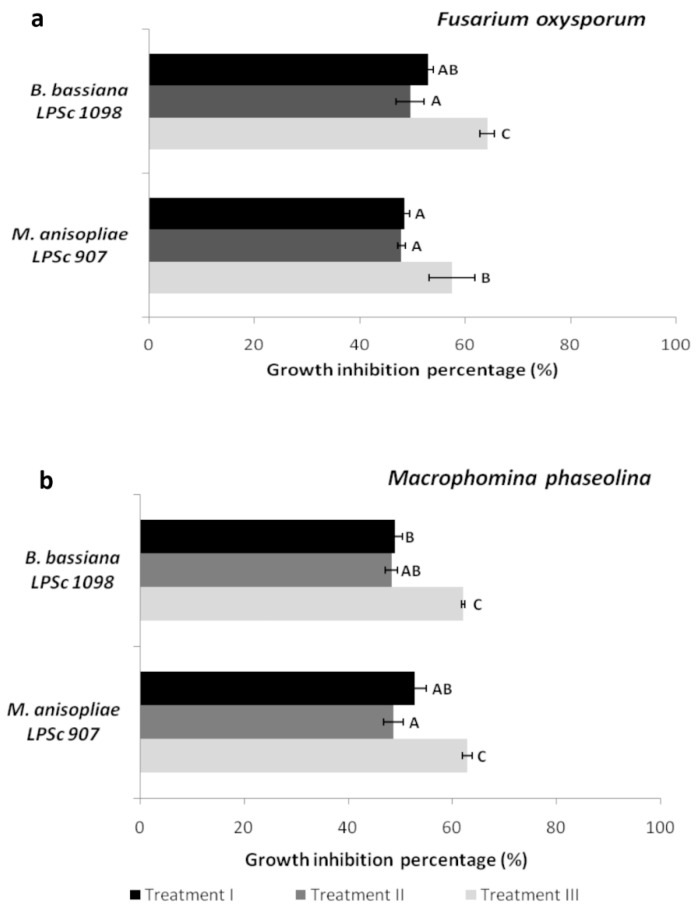
Inhibition percentage of radial growth of *F. oxysporum* (**a**) and *M. phaseolina* (**b**) by the fungal entomopathogens *B. bassiana* and *M. anisopliae* in the dual plate assay. Results are expressed as mean values (±SEM). Bars with different letters differ significantly at *p* < 0.05 (Tukey multiple range test after two-way ANOVA).

**Figure 5 jof-10-00093-f005:**
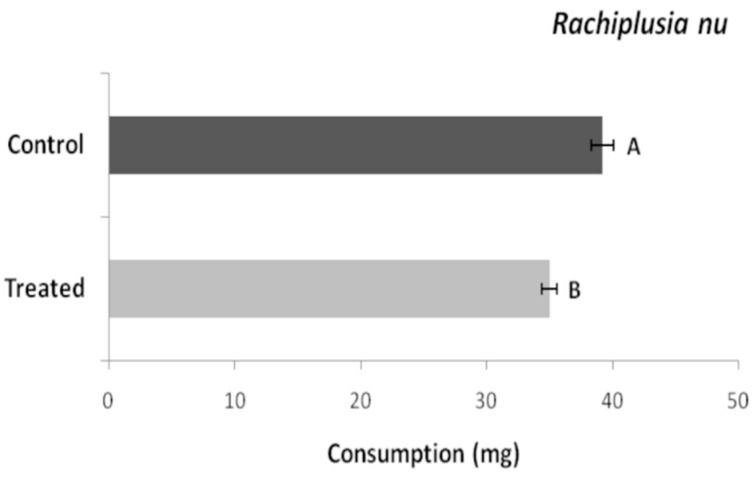
*Beauveria bassiana* colonized (treated) and non-colonized (control) leaves consumed (mg/insect) by *Rachiplusia nu* second instar larvae. Bars indicate mean values (±SEM). Bars with different letters differ significantly at *p* < 0.05 (Student’s *t* test).

**Table 1 jof-10-00093-t001:** Results of ANOVA for antagonist factor, treatment factor, and the interaction between both factors (antagonist × treatment).

	*F. oxysporum*			*M. phaseolina*		
	F	df	*p*	F	df	*p*
Antagonist	16.56	1	0.0016	11.97	1	0.0047
Treatment	50.58	2	<0.0001	158.66	2	<0.0001
Antagonist × Treatment	1.97	2	0.018	2.18	2	0.015

Significant at the *p* < 0.05 probability level.

## Data Availability

Data are contained within the article.

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
