# Peer review of "Entomopathogenic Fungi as Dual Control Agents against Two Phytopathogens and the Lepidopteran Pest Rachiplusia nu in Soybean (Glycine max (L.) Merr)"

_jof, 2024, doi:10.3390/jof10020093_

Round 1
Reviewer 1 Report
Over all manuscript is well written and well presented. I have no major problems with the manuscript, My comments are for improvement of the precision and clarity of wording. Some specific comments are on the hard copy:

Author Response
Authors appreciate reviewer comments that have improved the ms.
All comments made by the reviewer were incorporated in the ms and were highlighted in red.

Reviewer 2 Report
Dear authors,
The manuscript presents the effect of the entomopathogenic fungi Beauveria bassiana and Metarhizium anisopliae on the management of the phytopathogenic fungi Fusarium oxysporum and Macrophomina phaseolina and on the lepidopteran Rachiplusia nu. The results are interesting and very important in a context of integrated pest and disease management.
The introduction clearly presents what motivated the research, using literature and updating it. The topic material and methods are robust and widely described in the literature. The results are appropriate to the topic material and methods. The discussion is relevant and appropriate to the results.
In order to contribute to the text, I recommend that the authors revise the text to adapt the scientific names (example L. 114, the fungus B. bassiana is not in italics).
L. 79 replace “Methods” with “Material and methods”.
L 209-237 – Long text where data from other authors are presented instead of discussing the research results in a more appropriate way.
Author Response

(The authors gave the same response as above.)
